# Fused Triazole-Azepine Hybrids as Potential Non-Steroidal Antiinflammatory Agents

Sergii Demchenko [1], Roman Lesyk [2,*], Oleh Yadlovskyi [3], Serhii Holota [2,4], Sergii Yarmoluk [1], Sergii Tsyhankov [5] and Anatolii Demchenko [5]

1 Institute of Molecular Biology and Genetics of NASU, Zabolotnogo Str. 150, 03143 Kyiv, Ukraine; demcha.chem@gmail.com (S.D.); serhiy.yarmoluk@gmail.com (S.Y.)
2 Department of Pharmaceutical, Organic and Bioorganic Chemistry, Danylo Halytsky Lviv National Medical University, Pekarska 69, 79010 Lviv, Ukraine; golota_serg@yahoo.com
3 Institute of Pharmacology and Toxicology, Anton Tsedik 14, 03057 Kiev, Ukraine; yadlovskyi.oleg@gmail.com
4 Department of Organic Chemistry and Pharmacy, Lesya Ukrainka Volyn National University, Volya Avenue 13, 43025 Lutsk, Ukraine
5 Department of Chemistry and Pharmacy, Nizhyn Mykola Gogol State University, 16600 Nizhyn, Ukraine; vistss@gmail.com (S.T.); demch7758@ukr.net (A.D.)
* Correspondence: dr_r_lesyk@org.lviv.net; Tel.: +380-(32)-275-5966

**Abstract:** Non-steroidal anti-inflammatory drugs (NSAIDs) are one of the oldest and most widely used groups of drugs nowadays. However, the problem of searching for and creating new NSAIDs remains open, primarily due to the risks owing to their short- and long-term use. In this context, triazole-azepine hybrid molecules are attractive and prospective objects for the rational design of novel potential NSAIDs. In the present work studies of 3-aryl-6,7,8,9-tetrahydro-5*H*-[1,2,4]triazolo[4,3-*a*]azepines as potential non-steroidal anti-inflammatory agents are reported. Evaluation of drug-like properties for all tested triazole-azepine hybrids was performed in silico using SwissADME. The screening of analgesic and anti-inflammatory activities was performed in vivo using acid-induced writhing and carrageenin-induced hind paw oedema models in mice. Derivatives with activity levels more potent compared with reference drugs ketorolac and diclofenac sodium were identified. Preliminary SAR was performed based on the screening results.

**Keywords:** NSAIDs; triazole; azepine; hybrid molecules; analgesic activity; anti-inflammatory activity

## 1. Introduction

Non-steroidal anti-inflammatory drugs (NSAIDs) present a widely used and therapeutically important pharmacological group with a large number of individual and combined approved drugs [1–3]. These drugs belong to chemically different/unrelated compounds and possess common therapeutic features such as anti-inflammatory, analgesic, and antipyretic activity. The analgesic effect is a very important component of the pharmacological profile of NSAIDs and this group of drugs provides effective and generally safe options for mild to moderate pain [4]. At the same time, there exists an unmet need for the development of new non-steroidal anti-inflammatory agents as well as for optimization of the properties of the known NSAIDs focused on efficacy increasing and adverse effects decreasing [5,6].

Heterocyclic scaffolds play a leading role in the process of research and development of new potential NSAIDs [7,8]. Among the variety of heterocyclic compounds, the derivatives with triazole and azepine rings are attractive and prospective objects for the focused and rational design of novel molecules with pharmacological profiles characterized for NSAIDs [8–10]. Azepine-bearing derivative **A** (Figure 1) was reported in [11] as a potential agent with a high level of anti-inflammatory and analgesic activity and with selective inhibition of COX-2. Derivative **B** (Figure 1) with 1,2,4-triazole-3-thiol scaffold in the

molecule showed an equal level of anti-inflammatory activity compared with classic NSAID sodium diclofenac [12] in vivo on the carrageenin model. The anti-inflammatory activity of diarylsubstituted 1,2,4-triazoles with hydroxamic acid or N-hydroxyurea moieties in the molecules has been evaluated [13] and it was found that all the compounds of the series showed dual inhibitory activity in vitro toward COX-2/LOX-5 and the most active derivative **C** (Figure 1) was more potent than celecoxib in the xylene edema model in vivo.

**Figure 1.** Selected molecules with azepine and 1,2,4-triazole scaffolds that possess analgesic and anti-inflammatory activity; background and design for the present research.

Methods and strategies of molecular hybridization are convenient tools for achieving the desired properties of heterocyclic molecules which have been successfully applied to the above-mentioned types of heterocycles for obtaining potential pharmacological agents with antitumor activity [14–18].

In our previous studies, we used the above-mentioned approaches in the design of novel analgesics among fused triazole-azepine hybrid molecules [19–23]. As result, a series of hit-compounds (**D**–**F**, Figure 1) with high activity levels equal to ketorolac and diclofenac sodium were identified in the in vivo models ("hot-plate" and "acetic acid-induced writhing test"). Herein we present the novel results of our studies dedicated to the design of potential non-steroidal anti-inflammatory agents and report about the synthesis of new 3-aryl-6,7,8,9-tetrahydro-5*H*-[1,2,4]triazolo[4,3-*a*]azepines, evaluation of their drug-like properties in silico, analgesic and anti-inflammatory activities in vivo.

## 2. Materials and Methods

### 2.1. General Information

All materials were purchased from commercial sources and used without purification. Melting points were measured in open capillary tubes and are uncorrected. The elemental analyses (C, H, N) were performed using the Perkin–Elmer 2400 CHN analyzer (Perkin–Elmer, Norwalk, CT, USA) and were within 0.4% of the theoretical values. The $^1$H and $^{13}$C NMR spectra were recorded on a Bruker AVANCE-400 spectrometer (Bruker, Bremen, Germany). All spectra were recorded at room temperature, except where indicated otherwise, and were referenced internally to solvent reference frequencies. Chemical shifts (δ) are quoted in ppm, and coupling constants (*J*) are reported in Hz. LC–MS spectra were obtained on a Finnigan MAT INCOS-50 (Thermo Finnigan LLC, San Jose, CA, USA). The reaction mixture was monitored by thin layer chromatography (TLC) using commercial

glass-backed TLC plates (Merck Kieselgel 60 F254, Merck, Darmstadt, Germany). Solvents and reagents that are commercially available were used without further purification.

### 2.2. Synthesis and Characterization of Compounds

Starting compound **3** was prepared according to protocol described in [24]. Derivatives **6a–g** were synthesized following the protocol described in [25].

#### 2.2.1. General Procedure of Synthesis of 3-aryl-6,7,8,9-tetrahydro-5*H*-[1,2,4]triazolo[4,3-*a*]azepines **7a–g**

A mixture of 10 mmole of corresponding hydrazide **6a–g** and 10 mmole of 7-methoxy-3,4,5,6-tetrahydro-2H-azepine **3** was refluxed for 3 h in 100 mL of toluene. The obtained solution was left for 12 h at room temperature. Subsequently, obtained solid products of derivatives **7a–g** were collected by filtration, washed with benzene, and recrystallized from the appropriate solvent.

#### 2.2.2. Characterization of Compounds **7a–g**

3-Phenyl-6,7,8,9-tetrahydro-5*H*-[1,2,4]triazolo[4,3-*a*]azepine (**7a**). Yield 75%, mp 167–168 °C. $^1$H NMR (400 MHz, DMSO-$d_6$, δ): 1.65–1.71 (m, 4H, 7,8-CH$_2$CH$_2$-), 1.75–1.81 (m, 2H, 6-CH$_2$), 2.96–3.00 (m, 2H, 9-CH$_2$), 3.97–4.00 (m, 2H, 5-CH$_2$), 7.50–7.54 (m, 5H, arom.). $^{13}$C NMR (100 MHz, DMSO-$d_6$, δ): 25.1, 25.9, 27.9, 29.7, 45.1, 127.5, 128.8, 128.9, 129.6, 153.8, 157.3. LCMS (ESI+) *m/z* 214.2 (100%, [M+H]$^+$). Anal. calc. for C$_{13}$H$_{15}$N$_3$: C 73.21%, H 7.09%, N 19.70%. Found: C 73.40%, H 7.20%, N 19.90%.

2-(6,7,8,9-Tetrahydro-5*H*-[1,2,4]triazolo[4,3-*a*]azepine-3-yl)-phenol (**7b**). Yield 73%, mp 242–243 °C. $^1$H NMR (400 MHz, DMSO-$d_6$, δ): 1.65–1.69 (m, 4H, 7,8-CH$_2$CH$_2$-), 1.78–1.81 (m, 2H, 6-CH$_2$), 2.94–2.98 (m, 2H, 9-CH$_2$), 3.76–3.80 (m, 2H, 5-CH$_2$), 6.80 (t, 1H, J = 7.4 Hz, arom), 7.00 (d, 1H, J = 7.8 Hz, arom), 7.30 (d, 1H, J = 7.8 Hz, arom), 7.40 (t, 1H, J = 7.4 Hz, arom), 10.1 (s, 1H, OH). $^{13}$C NMR (100 MHz, DMSO-$d_6$, δ): 25.1, 25.9, 27.9, 29.7, 45.0, 115.6, 116.6, 119.4, 128.6, 129.9, 153.8, 157.2, 157.5. LCMS (ESI+) *m/z* 230.2 (100%, [M+H]$^+$). Anal. calc. for C$_{13}$H$_{15}$N$_3$O: C 68.10%, H 6.59%, N 18.33%. Found: C 68.30%, H 6.80%, N 18.50%.

3-(6,7,8,9-Tetrahydro-5*H*-[1,2,4]triazolo[4,3-*a*]azepine-3-yl)-phenol (**7c**). Yield 70%, mp 239–240 °C. $^1$H NMR (400 MHz, DMSO-$d_6$, δ): 1.64–1.70 (m, 4H, 7,8-CH$_2$CH$_2$-), 1.75–1.79 (m, 2H, 6-CH$_2$), 2.90–2.94 (m, 2H, 9-CH$_2$), 3.96–3.99 (m, 2H, 5-CH$_2$), 6.90–6.95 (m, 3H, arom), 7.33 t, 1H, J = 7.3 Hz, arom), 9.84 (s, 1H, OH). $^{13}$C NMR (100 MHz, DMSO-$d_6$, δ): 25.1, 25.9, 27.9, 29.7, 45.0, 115.6, 116.6, 119.4, 128.6, 129.9, 153.8, 157.2, 157.5. LCMS (ESI+) *m/z* 230.2 (100%, [M+H]$^+$). Anal. calc. for C$_{13}$H$_{15}$N$_3$O: C 68.10%, H 6.59%, N 18.33%. Found: C 68.20%, H 6.70%, N 18.40%.

3-(4-Methoxyphenyl)-6,7,8,9-tetrahydro-5*H*-[1,2,4]triazolo[4,3-*a*]azepine (**7d**). Yield 77%, mp 167–168°C. $^1$H NMR (400 MHz, DMSO-$d_6$, δ): 1.64–1.69 (m, 4H, 7,8-CH$_2$CH$_2$-), 1.79–1.81 (m, 2H, 6-CH$_2$), 2.90–2.93 (m, 2H, 9-CH$_2$), 3.82 (s, 3H, OCH$_3$), 3.95–3.99 (m, 2H, 5-CH$_2$), 7.09 (d, 2H, J = 8.3 Hz, arom), 7.46 (d, 2H, J = 8.3 Hz, arom). $^{13}$C NMR (100 MHz, DMSO-$d_6$, δ): 25.2, 25.9, 27.9, 29.8, 45.0, 55.3, 114.2, 119.7, 130.4, 153.7, 157.0, 160.1. LCMS (ESI+) *m/z* 244.2 (100%, [M+H]$^+$). Anal. calc. for C$_{14}$H$_{17}$N$_3$O: C 69.11%, H 7.04%, N 17.27%. Found: C 69.30%, H 7.20%, N 17.40%.

3-(3,4,5-Trimethoxyphenyl)-6,7,8,9-tetrahydro-5*H*-[1,2,4]triazolo[4,3-*a*]azepine (**7e**). Yield 71%, mp 116–117 °C. $^1$H NMR (400 MHz, DMSO-$d_6$, δ): 1.64–1.68 (m, 4H, 7,8-CH$_2$CH$_2$-), 1.79–1.82 (m, 2H, 6-CH$_2$), 2.90–2.94 (m, 2H, 9-CH$_2$), 3.72 (s, 3H, OCH$_3$), 3.83 (s, 6H, 2*OCH$_3$), 3.99–4.03 (m, 2H, 5-CH$_2$), 6.79 (s, 2H, arom). $^{13}$C NMR (100 MHz, DMSO-$d_6$, δ): 25.2, 25.9, 27.8, 29.7, 45.1, 56.0, 60.1, 106.4, 122.8, 138.4, 153.0, 153.9, 157.1. LCMS (ESI+) *m/z* 304.0 (100%, [M+H]$^+$). Anal. calc. for C$_{16}$H$_{21}$N$_3$O$_3$: C 63.35%, H 6.98%, N 13.85%. Found: C 63.60%, H 7.00%, N 14.00%.

3-p-Tolyl-6,7,8,9-tetrahydro-5*H*-[1,2,4]triazolo[4,3-*a*]azepine (**7f**). Yield 77%, mp 157–158 °C. $^1$H NMR (400 MHz, DMSO-$d_6$, δ): 1.64–1.69 (m, 4H, 7,8-CH$_2$CH$_2$-), 1.78–1.81 (m, 2H, 6-CH$_2$), 2.38 (s, 3H, CH$_3$), 2.91–2.94 (m, 2H, 9-CH$_2$), 3.98–4.01 (m, 2H, 5-CH$_2$), 7.35 (d, 2H, J = 8.0 Hz, arom), 7.42 (d, 2H, J = 8.0 Hz, arom). $^{13}$C NMR (100 MHz, DMSO-$d_6$, δ):

20.9, 25.2, 25.9, 27.9, 29.7, 45.0, 124.7, 128.8, 129.4, 139.2, 153.8, 157.1. LCMS (ESI+) *m/z* 304.0 (100%, [M+H]$^+$). Anal. calc. for $C_{14}H_{17}N_3$: C 73.98%, H 7.54%, N 18.49%. Found: C 74.10%, H 7.70%, N 18.60%.

N-[4-(6,7,8,9-Tetrahydro-5*H*-[1,2,4]triazolo[4,3-*a*]azepin-3-yl-phenyl]acetamide (**7g**). Yield 81%, mp 219–220 °C. $^1$H NMR (400 MHz, DMSO-*d*$_6$, δ): 1.64–1.70 (m, 4H, 7,8-CH$_2$CH$_2$-), 1.78–1.81 (m, 2H, 6-CH$_2$), 2.08 (s, 3H, CH$_3$), 2.91–2.94 (m, 2H, 9-CH$_2$), 3.97–4.01 (m, 2H, 5-CH$_2$), 7.46 (d, 2H, J = 8.4 Hz, arom), 7.74 (d, 2H, J = 8.4 Hz, arom), 10.2 (s, 1H, NH). $^{13}$C NMR (100 MHz, DMSO-*d*$_6$, δ): 24.0, 25.2, 25.9, 27.9, 29.7, 45.0, 118.9, 121.8, 129.4, 140.1, 153.7, 157.1, 168.6. LCMS (ESI+) *m/z* 271.0 (100%, [M+H]$^+$). Anal. calc. for $C_{15}H_{18}N_4O$: C 66.64%, H 6.71%, N 20.73%. Found: C 66.80%, H 6.90%, N 20.90%.

*2.3. Pharmacology Studies*

2.3.1. Animals

Female non-linear mice (18–22 g) were used for the experimental studies. The animals were housed in a quarantine facility for 7 days before the experiment was started. Throughout the experiment, the animals were randomized in groups (n = 6) per cage with bedding composed of wood shavings (exchanged daily). The animals had free access to a standard commercial diet and water. The animals were kept under a stable regimen of 12 h light/12 h darkness. The animals were treated humanely throughout the study period adhering to the guideline for the use and care of animals in the Declaration of Helsinki (National Research Council, 2011). The experiment design and study protocol were approved by the Animal Ethics Committee of the Institute of Pharmacology and Toxicology of the National Academy of Medical Sciences of Ukraine, protocol No. 14, 20 June 2022.

2.3.2. Analgesic Activity

The acetic acid-induced writhing model in mice was used for studying the analgesic activity of synthesized compounds. Compounds **7a–g** were administered once orally (p.o.) at the dose of 25 mg/kg [26] in the form of an aqueous-ethanol emulsion using Twin-80 as an emulgator. Ketorolac (Ketorolac tromethamine (JSC "Lek-Chem", Ukraine), was administered once orally (p.o.) in the form of an aqueous solution at the dose of 25 mg/kg. Tested compounds and reference drug were administrated 60 min before the administration of 0.6% acetic acid solution. The number of writhing behaviors produced were determined in each group for the following 10 min [27]. Inhibition of writhing was calculated by the formula below and compared with the standard drug (ketorolac):

$$\text{Inhibition of writhing, \%} = \{(Wc - Wt) \times 100\%\}/Wc$$

where, Wc = number of writhing in the control group; Wt = number of writhing of the experimental group.

2.3.3. Anti-Inflammatory (Antiexudative) Activity

The carrageenin-induced hind paw oedema was produced by the method of Winter et al. [28]. The synthesized compounds were intraperitoneally injected in a dose 25 mg/kg (in saline solution with one drop of Tween-80™). Diclofenac (tablets "Diclofenac sodium", "Zdorovja narodu", Ukraine) in dose 25 mg/kg was used as reference drug. The antiexudative activity (inflammation inhibition) was expressed as a decrease of rats-paw oedema, was calculated using the equation and was given in percentage:

$$\text{Inhibition, \%} = (\Delta Vcontrol - \Delta Vexperiment)/\Delta Vcontrol \times 100\%$$

where, ΔVcontrol and ΔVexperiment—the mean values of the volume difference for control and experimental animals hinds respectively.

## 3. Results and Discussion

### 3.1. Synthesis and Characterization of Derivatives *7a–g*

Commercially available azepan-2-one (caprolactam) **1** and aromatic carboxylic acids **4a–g** were used as initial compounds for the construction of the target derivatives **7a–g**. Azepan-2-one **1** was transformed to the 7-methoxy-3,4,5,6-tetrahydro-2H-azepine **3** following the protocol reported in [24] (Scheme 1A), whereas the routine scheme including esterification with the next hydrazinolysis by hydrazine hydrate of corresponding acids **4a–g** was used for obtaining hydrazides **6a–g** (Scheme 1B).

**R= (a) H; (b) 2-OH; (c) 3-OH; (d) 4-CH$_3$O; (e) 3,4,5-(CH$_3$O)$_3$; (f) 4-CH$_3$; (g) 4-NHCOCH$_3$**

**Scheme 1.** Synthetic routes for the preparation of initial building blocks **3** and **6a–g**. Reagents and conditions: (**A**) **1** (10 mmole), (CH$_3$)$_2$SO$_4$ (10 mmole), benzene, stirring 3 h, 60 °C; **2** (10 mmole), K$_2$CO$_3$ (20 mmole), H$_2$O, stirring 1 h, 5 °C; (**B**) **4a–g** (10 mmole), ethanol (20 mL), H$_2$SO$_4$ concentrated (one drop), reflux 3 h; **5a–g** (10 mmole), hydrazine hydrate (11 mmole), ethanol (20 mL).

Physicochemical properties and spectral characteristics of the synthesized derivatives **6a–g** correspond to literature data [29–32].

Targeting triazole-azepine hybrids **7a–g** were obtained with a yield of 70–81% via the condensation of **3** with hydrazides **6a–g** and subsequent cyclization of the intermediate products following the method reported in [25] (Scheme 2).

**R= (a) H; (b) 2-OH; (c) 3-OH; (d) 4-CH$_3$O; (e) 3,4,5-(CH$_3$O)$_3$; (f) 4-CH$_3$; (g) 4-CH$_3$CONH**

**Scheme 2.** Synthesis of triazole-azepine hybrids **7a–g**. Reagents and conditions: *i*—**3** (10 mmole), **6a–g** (10 mmole), toluene (100 mL), reflux 3 h, r.t. 12 h.

The structure of synthesized hybrid molecules **7a–g** was confirmed using $^1$H, $^{13}$C NMR and LC-MS spectra (copies of spectra are presented in the Supplementary Materials). In the $^1$H NMR spectra of compounds **7a–g** protons of the azepine ring give a complex

pattern with four multiplets at ~1.65–1.70, ~1.81–1.84, ~2.93–2.96, and ~3.80–4.03 ppm. The protons of OH-groups of compounds **7b** and **7c** were resonated as singlets at 10.10 and 9.84 ppm, respectively. The molecular ion peaks observed in the mass spectra for the *m/z* values of the synthesized compounds in the positive ionization mode corresponded to Mr+1 which confirmed the formation of the derivatives **7a–g**.

### 3.2. *In Silico Evaluation of Drug-Likeness Parameters and Pharmacokinetics Properties of Compounds 7a–g Using the SwissAdme*

The series of drug-likeness and pharmacokinetics properties of the compounds **7a–g** were evaluated in silico using the SwissAdme of the Swiss Institute of Bioinformatics website [33]. The calculated prognostic data are highlighted in Table (Table 1).

**Table 1.** Drug-likeness and pharmacokinetics properties of derivatives **7a–g** calculated in silico.

| Compound/Parameter | Lipinski Rules | | | | Veber Rules | | Fraction Csp3 $\geq 0.25$ | GI Absorption | BBB Permeant | P-gp Substrate |
|---|---|---|---|---|---|---|---|---|---|---|
| | MW $\leq 500$ | Log P $\leq 5$ | NHD $\leq 5$ | NHA $\leq 10$ | NBR $\leq 10$ | TPSA $\leq 140$ | | | | |
| 7a | 213.28 | 1.96 | 0 | 2 | 1 | 30.71 | 0.38 | High | Yes | Yes |
| 7b | 229.28 | 2.10 | 1 | 3 | 1 | 50.94 | 0.38 | High | Yes | Yes |
| 7c | 229.28 | 2.04 | 1 | 3 | 1 | 50.94 | 0.38 | High | Yes | Yes |
| 7d | 243.30 | 2.45 | 0 | 3 | 2 | 39.94 | 0.43 | High | Yes | Yes |
| 7e | 303.36 | 2.43 | 0 | 5 | 4 | 58.40 | 0.50 | High | Yes | Yes |
| 7f | 227.30 | 2.79 | 0 | 2 | 1 | 30.71 | 0.43 | High | Yes | Yes |
| 7g | 270.33 | 2.04 | 1 | 3 | 3 | 59.81 | 0.40 | High | Yes | Yes |

GI—gastrointestinal; BBB—blood-brain barrier; P-gp—P-glycoprotein1.

Accordingly, with obtained data, all derivatives **7a–g** correspond to the Lipinski and Veber rules. Moreover, all the molecules possess predicted satisfactory pharmacokinetic parameters such as a high level of gastrointestinal absorption, the ability to pass through the blood-brain barrier and to be a substrate for P-glycoprotein1. Additionally, should be noted that calculated by the SwissAdme fraction of Csp$^3$ for all the molecules were in the range from 0.38 to 0.50.

### 3.3. *In Vivo Studies of Analgesic and Anti-Inflammatory Activity of Compounds 7a–g and Emprical SAR*

All the derivatives **7a–g** were tested in vivo for their analgesic and anti-inflammatory activity using the acetic acid-induced writhing model and carrageenin-induced inflammation models in mice, respectively. Ketorolac and diclofenac sodium, nonselective inhibitors of both cyclooxygenase-1 (COX-1) and cyclooxygenase-2 (COX-2), were used as a reference drugs in the experiments. The study results are presented in Figure 2A,B.

Analgesic activity screening results revealed that among derivatives **7a–g** only two compounds (**7b** and **7d**) showed significant activity levels compared with the reference drug (Figure 2A). Derivative **7b** was more potent than ketorolac in the experimental conditions with inhibition of writhing value 95.5% compared with the 85.9% value for ketorolac. Compound **7d** possesses lower analgesic activity with inhibition of writhing value 76.6%. Synthesized hybrid molecules **7a,c,e,g** were characterized with inhibition of writhing values in the range of 17.1%–45.9%, whereas for the derivative **7f** analgesic activity was not observed at all.

Anti-inflammatory activity screening results revealed that tested derivatives **7a–g** possess promising anti-exudative effects, and all the compounds were more potent than the reference drug diclofenac sodium (Figure 2B). The inflammation inhibition index for compounds **7a–g** was in the range of 50.3%–73.0%, whereas for the reference drug diclofenac sodium it was 44.2% in the experimental conditions.

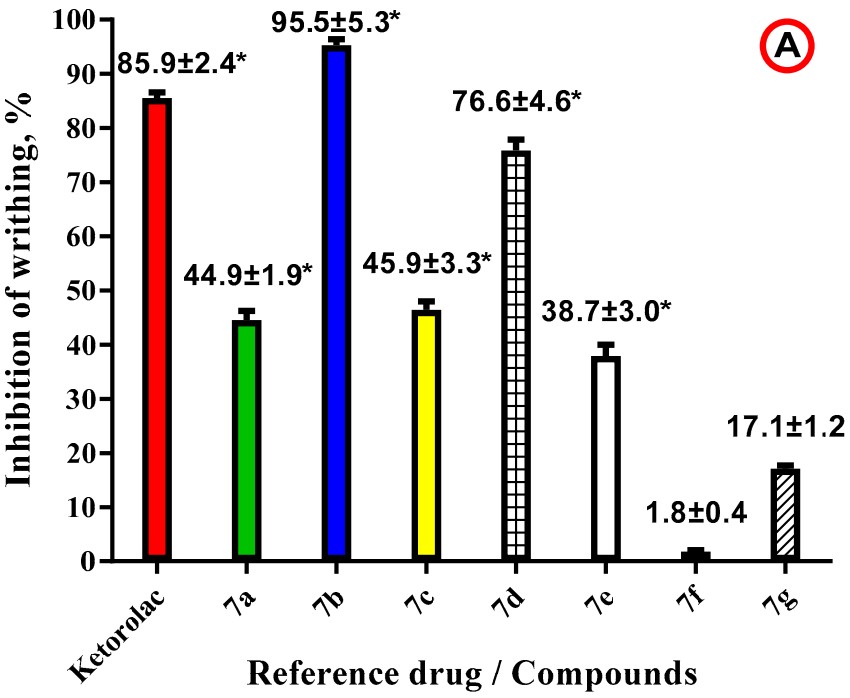

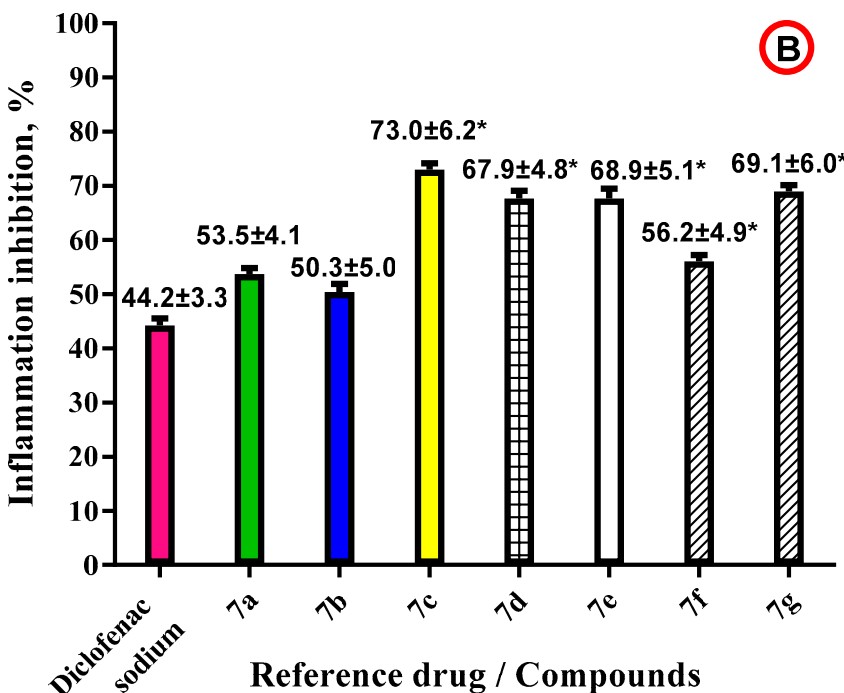

**Figure 2.** Pharmacological screening results for compounds **7a–g** in vivo in mice. (**A**) Analgesic activity, acetic acid-induced writhing model (intraperitoneally use; doses: 0.6% acetic acid; ketorolac—25 mg/kg). (**B**) Anti-inflammatory activity, carrageenin-induced paw edema model (intraperitoneally use; doses: carrageenin 1%, 0.1 mL; diclofenac sodium—8 mg/kg). Tested compounds **7a–g** were used in doses 25 mg/kg in both models. All the data presented as M ± m; n = 6 in each group. * $p < 0.05$ compared to the control group.

From the point of view of structure–activity relationships (SAR), it is worth noting that the derivatives with a hydroxyl group in the phenyl ring (**7b**, **7c**) were the most active both in the case of analgesic and anti-inflammatory activity (Figure 3). The change of

OH-group position from ortho (**7b**) to meta (**7c**) leads to a significant decrease in analgesic activity level compared with the ketorolac effect. Whereas the same structural change in the case of anti-inflammatory activity keeps the effect at a more potent level compared with the reference drug. Also, in the context of SAR, should be noted that the presence of methoxy-group (**7d**) contributed to the significant analgesic activity, whereas the presence of other substituents does not lead to the appearance of an analgesic effect. Summarizing the pharmacological screening results, compound **7b** is of interest for further development and optimization as a potential non-steroidal anti-inflammatory agent.

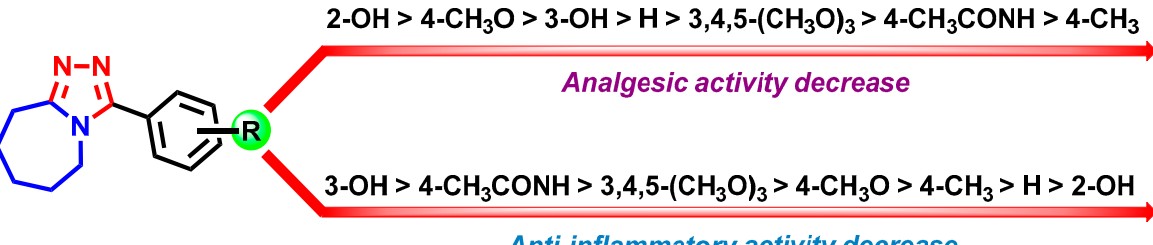

**Figure 3.** Structure—analgesic and anti-inflammatory activity relationships for synthesized triazole-azepine hybrid molecules **7a–g**.

## 4. Conclusions

In the present work, the evaluation of triazole-azepine hybrid molecules as potential NSAIDs is reported. All the studied hybrid molecules correspond to key drug-likeness parameters and possess predicted satisfactory pharmacokinetic properties according to the in silico evaluation performed using SwissADME. Analgesic and anti-inflammatory activities were studied in vivo using acid-induced writhing and carrageenin model of inflammatory oedema on white mice for the synthesized hybrids. Highly active derivatives with a promising effect that exceeds the activity level of reference drugs ketorolac and sodium diclofenac were identified. According to the obtained results, compound **7b** represents an interest for in-depth studies as a potential non-steroidal anti-inflammatory agent. Preliminary SARs contributing to the medicinal chemistry of triazole and azepine derivatives and their hybrid molecules are presented. The polypharmacological profile of studied hybrids is an argument for in-depth studies of these types of molecules applying classic pharmacological as well as computational chemistry methods with the aim of identifying their potential molecular targets.

**Supplementary Materials:** The following supporting information can be downloaded at: https://www.mdpi.com/article/10.3390/scipharm91020026/s1, Figures S1–S21: [1]H NMR, [13]C NMR, and LC–MS spectra of compounds **7a–g**.

**Author Contributions:** Conceptualization, S.D., R.L. and A.D.; methodology, S.D., O.Y., S.Y. and S.T.; software, S.H.; validation, S.D., O.Y., S.Y. and S.T.; formal analysis, S.D., O.Y., S.H. and S.T.; investigation, S.D., O.Y. and S.T.; resources, S.D., R.L., O.Y., S.Y. and A.D.; data curation, S.D., R.L., O.Y., S.H., S.Y., S.T. and A.D.; writing—original draft preparation, S.D., R.L., O.Y., S.H., S.Y., S.T. and A.D.; writing—review and editing, S.D., R.L., O.Y., S.H., S.Y., S.T. and A.D.; visualization, S.D., R.L. and A.D.; supervision, S.D., R.L. and A.D.; project administration, S.D., R.L. and A.D.; funding acquisition, S.D., R.L. and A.D. All authors have read and agreed to the published version of the manuscript.

**Funding:** The authors sincerely thank the Krzysztof Skubiszewski Foundation for financial support of this study.

**Institutional Review Board Statement:** Not applicable.

**Informed Consent Statement:** Not applicable.

**Data Availability Statement:** The data presented in this study are available in this article.

**Acknowledgments:** The authors would like to thank all the brave defenders of Ukraine who made the finalization of this article possible.

**Conflicts of Interest:** The authors declare no conflict of interest.

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
