# Peer review of "Fused Triazole-Azepine Hybrids as Potential Non-Steroidal Antiinflammatory Agents"

_scipharm, doi:10.3390/scipharm91020026_

Round 1

Reviewer 1 Report

The authors selected a series of compounds to test the analgesic activity. THey have clearly described the thought process of how they have selected the compounds for evaluation. They gave reference to the synthetic method, however a full description of the synthetic procedure would have been more helpful in the materials and methods section. The authors clearly described the mouse model study and effectively presented the research outcomes. They have also described the structure activity relationship of the selected compounds with the pharmacological potency. The manuscript is recommended for publication.

Author Response

Dear reviewer!
Many thanks for Your time spending and efforts in reviewing the manuscript. We appreciate your positive evaluation of our work.

Reviewer 2 Report

The manuscript by Demchenko et al., describes the design, synthesis and biological evaluation of a small library of 3-aryl-6,7,8,9-tetrahydro-5H-[1,2,4]triazolo[4,3-a]azepines as potential non-steroidal anti-inflammatory agents. For all the synthesized derivatives in silico evaluation of their drug-like properties was performed using SwissADME. The screening of analgesic and anti-inflammatory activities was performed in vivo using acid-induced writhing and carrageen-in-induced hind paw oedema models in mice. In vivo studies of analgesic and anti-inflammatory activity of the novel compounds were performed, evidencing highly active derivatives with a promising effect that exceeds the activity level of the reference drugs, namely ketorolac and sodium diclofenac. In addition, preliminary SARs contributing to the medicinal chemistry of triazole and azepine derivatives and their hybrid molecules are presented. Of note, summarizing the pharmacological screening results, compound 7b proves to be the most interesting compound for further development and optimization as a potential non-steroidal anti-inflammatory agent.

The paper is quite well written and appropriately organized. Therefore, it is suitable to be published after the following Minor Revisions:

-          Introduction, p. 2, line 55. Please, put in vitro in italic.

-          Introduction, p. 2, line 59. Please remove “are” or “could be” depending on the sense the sentence must assume.

-          Results and Discussion. Please add the reagents and conditions of the synthesis to scheme 1.

-          Results and Discussion, p. 8, line 282. Please replace doesn’t with does not.

Author Response

Dear reviewer!

Many thanks for Your time spending and efforts in reviewing the manuscript.

We would like to comment on the main points.

Introduction, p. 2, line 55. Please, put in vitro in italic.

            Corrected as suggested

Introduction, p. 2, line 59. Please remove “are” or “could be” depending on the sense the sentence must assume.

            Corrected as suggested  

Results and Discussion. Please add the reagents and conditions of the synthesis to scheme 1.

            Corrected as suggested. The reagents and conditions of the synthesis to scheme 1 were added.

Results and Discussion, p. 8, line 282. Please replace doesn’t with does not.

            Corrected as suggested

Reviewer 3 Report

Dear Editor,

I have carefully reviewed the manuscript which has many critical issues. The work looks good and is also well written but the molecules are, with a few exceptions, completely known for their synthesis and their antitumor activity .... which is not referred to in the work. Therefore I don't understand what the authors want to do..... describe an activity parallel to the anticancer one??? so they don't have to do chemical but pharmacological work ..... It is not possible to describe the anti-inflammatory and analgesic activity regardless of an anti-tumor activity or in any case the latter should have been referred to in the new work ..... It is not possible to report an entire chemical part thinking or making people think of an innovative scaffold for the pharmacological activity described. Therefore, on the basis of these considerations, I believe that the manuscript is not suitable for publication.... and if the authors want to describe this pharmacological activity they must write a pharmacological manuscript also referring to the anti-tumor activity previously described for these molecules

The synthesis and the study of the antitumor activity of 1,4-diaryl-5,6,7,8-tetrahydro-2,2a,8a triazacyclopenta[cd]azulene derivatives

By: Demchenko, S. A.; Fedchenkova, Yu. A.; Bobkova, L. S.; Artemchuk, L. P.; Demchenko, A. M.

Zhurnal Organichnoi ta Farmatsevtichnoi Khimii (2019), 17, (1), 3-12. Publisher: (NatsionalÏnii Farmatsevtichnii Universitet, ) CODEN:ZOFKAM ISSN:2518-1548

Derivatives of 1,4-diaryl-5,6,7,8-tetrahydro-2,2a,8a-triazacyclopenta[c,d]azulene

By: Demchenko, A. M.

Chemistry of Heterocyclic Compounds (New York)(Translation of Khimiya Geterotsiklicheskikh Soedinenii) (2001), 36, (8), 985-988. Publisher: (Consultants Bureau, ) CODEN:CHCCAL ISSN:0009-3122.

Regards

Author Response

Dear reviewer!

Many thanks for Your revision and constructive comments that helped improve the manuscript.

We would like to comment on the main points.

I have carefully reviewed the manuscript which has many critical issues. The work looks good and is also well written but the molecules are, with a few exceptions, completely known for their synthesis and their antitumor activity .... which is not referred to in the work. Therefore I don't understand what the authors want to do..... describe an activity parallel to the anticancer one??? so they don't have to do chemical but pharmacological work ..... It is not possible to describe the anti-inflammatory and analgesic activity regardless of an anti-tumor activity or in any case the latter should have been referred to in the new work ..... It is not possible to report an entire chemical part thinking or making people think of an innovative scaffold for the pharmacological activity described. Therefore, on the basis of these considerations, I believe that the manuscript is not suitable for publication.... and if the authors want to describe this pharmacological activity they must write a pharmacological manuscript also referring to the anti-tumor activity previously described for these molecules

We take into account your suggestion and added the mentioned articles to the reference list (17. Chem. Heterocycl. Compd. 2000, 36, 985–988. https://doi.org/10.1007/BF02256986; 18. J. Org. Pharm. Chem. 2019, 17, 3-12. https://doi.org/10.24959/ophcj.19.959). At the same time, we would like to note that the compounds described in our work are derivatives of 3-aryl-6,7,8,9-tetrahydro-5H-[1,2,4]triazolo[4,3-a]azepines, for which antitumor activity was not studied. The specified compounds are only starting materials for the synthesis of 1,4-diaryl-5,6,7,8-tetrahydro-2,2a,8a-triazacyclopenta[cd]azulene derivatives, which have shown antitumor effects. These explanations are also given in the text of the manuscript in the Introduction section.

Round 2

Reviewer 3 Report

In my opinion it is still not suitable for publication. The work can only be published as a pharmacological work

Author Response

Dear reviewer!

Many thanks for Your revision. Some changes have been incorporated in the revised manuscript (green highlight)

We would like to comment on the main point.

In my opinion it is still not suitable for publication. The work can only be published as a pharmacological work

We have put the pharmacological activity studies in the foreground rather than the synthesis.

Yours faithfully

Roman Lesyk